# ABC Transporters in *Prorocentrum lima* and Their Expression Under Different Environmental Conditions Including Okadaic Acid Production

**DOI:** 10.3390/md17050259

**Published:** 2019-04-30

**Authors:** Song Gu, Shao-Wen Xiao, Jian-Wei Zheng, Hong-Ye Li, Jie-Sheng Liu, Wei-Dong Yang

**Affiliations:** Key Laboratory of Eutrophication and Red Tide Prevention of Guangdong Higher Education Institutes, College of Life Science and Technology, Jinan University, Guangzhou 510632, China; koosle@163.com (S.G.); swxjbl@126.com (S.-W.X.); jwzheng89@126.com (J.-W.Z.); thyli@jnu.edu.cn (H.-Y.L.); tjsliu@jnu.edu.cn (J.-S.L.)

**Keywords:** ABC transporters, *Prorocentrum lima*, okadaic acid

## Abstract

*Prorocentrum lima* is a typical benthic toxic dinoflagellate, which can produce phycotoxins such as okadaic acid (OA). In this study, we identified three ABC transporter genes (*ABCB1*, *ABCC1* and *ABCG2*) and characterized their expression patterns, as well as OA production under different environmental conditions in *P. lima*. We found that the three ABC transporters all showed high identity with related ABC proteins from other species, and contained classical features of ABC transport proteins. Among them, *ABCG2* was a half size transporter. The three ABC transporter genes displayed various expression profiles under different conditions. The high concentration of Cu^2+^ could up-regulate *ABCB1*, *ABCC1* and *ABCG2* transcripts in *P. lima*, suggesting the potential defensive role of ABC transporters against metal ions in surrounding waters. Cu^2+^, in some concentration, could induce OA production; meanwhile, tributyltin inhibited OA accumulation. The grazer *Artemia salina* could induce OA production, and *P. lima* displayed some toxicity to the grazer, indicating the possibility of OA as an anti-grazing chemical. Collectively, our results revealed intriguing data about OA production and the expression patterns of three ABC transporter genes. However, we could not find any significant correlation between OA production and expression pattern of the three ABC transporters in *P. lima*. Our results might provide new molecular insights on the defensive responses of *P. lima* to the surrounding environment.

## 1. Introduction

ATP-binding cassette transporters (ABC) are a large family of membrane proteins with a variety of functions, which exist in almost all organisms [1]. The most common function of ABC transporters is to move multiple substrates into or out of the cytoplasm, such as ions, sugars, amino acids, peptides, toxic metabolites and heterotrophic substances, as well as drugs and toxins [2]. A typical structure of eukaryotic ABC transporters consists of two conserved structural domains: a transmembrane domain (TMD) and a nucleotide binding domain (NBD) [3]. Based on the protein solubility, TMDs existence, functions and amino acid sequences, ABC transporters have been assigned into eight subfamilies including full size and half size transporters: A, B, C, D, E, F, G, and I [4]. Generally, ABC transporters in plants are engaged in secondary metabolite transport, heavy metal detoxification, antibiotic transport and phytohormone transport, etc. [5]. Among them, pleiotropic drug resistance (PDR, ABCG), multidrug resistance (MDR, P-glycoprotein, ABCB), and multidrug resistance associated protein (MRP, ABCC) are the well-characterized proteins responsible for chemo-resistance and self-toxicity [6,7,8]. Therefore, recent studies mainly focused on these three subfamilies. For examples, *AtMRP6* expression was significantly up-regulated in *Arabidopsis thaliana* under cadmium stress, and plasma membrane-localized AtPDR8 has been shown to have the capability of mediating Cd^2+^ efflux, thereby playing a role in cadmium resistance [9,10].

To date, there are few reports about ABC transporters in algae, and their functions are not well understood. Schulz and Kolukisaoglu, for the first time, proposed that the *Chlamydomonas* genome consists of about 100 ORFs with domains or proteins similar to ABC transporters [11]. Later on, Merchant et al. put forward that *Chlorella* contained similar numbers of ABC transporter families and total ABC transporter genes as a terrestrial plant [12]. However, only 26 kinds of ABC transporters were found in *Ceratophyllum demersum*, which might contribute to its simpler transport systems, since most *C. demersum* cells have direct access to nutrients in surrounding waters [13]. It was suggested that, in *Chlamydomonas*, an ABC transporter was involved in lipid transport between endoplasmic reticulum and chloroplast, as well as in cadmium tolerance [14,15]. In the case of microalgae, Scherer et al., for the first time, confirmed the presence of MRP in diatoms using MK571, a specific inhibitor of MRP [16], while Hou et al. provided evidence for the *MRP* transporter in the plasma membrane of dinoflagellate [17]. Using gene chip technology, Yang et al. found several ABC transporter genes in the dinoflagellate *Alexandrium minutum* [18]. Based on the proteomic analysis, Wang et al. found a significantly up-regulated ABC transporter related protein in strain ACHK-NT, a toxicity-lost mutant of *Alexandrium catenella* [19]. Interestingly, Carvalho et al. found that expression levels of ABC transporter genes were significantly up-regulated by benzopyrene in *Thalassiosira pseudonana* [20], indicating the potential role of ABC transporters defense against environmental pollutants. In aquatic ecosystems, marine microalgae are often exposed to water containing exogenous toxins and harmful substances. Thus, microalgae should have a similar protection or defense system against harmful or cytotoxic substances. Taken together with the roles of ABC transporters in algae tolerance to heavy metals [14,21] and in the microcystin production in *Microcystis aeruginosa* PCC 7806 [22,23], it is reasonable to speculate that ABC transporters might have important roles in the transport or sequestration of endogenous secondary metabolites and xenobiotic pollutants. In fact, it has been proposed that ABC transporters export polysaccharides outside of dinoflagellate cells [24]. Nevertheless, to date there is no study available concerning the gene structure and function of ABC transporters in dinoflagellate, despite their biological significance.

*Prorocentrum lima* is a cosmopolitan epiphytic-benthic toxic dinoflagellate, which often attaches to sand, seaweed, benthic debris and coral reefs on the surface of algae [25]. It has been found that *P. lima* can produce phycotoxins such as okadaic acid (OA), dinophysistoxin-1 (DTX1), DTX2, and their derivatives, and in turn, associated with DSP episodes in different parts of the world [26]. Tremendous progress towards identifying and assessing toxic components from *P. lima* have been made, and potential ecological function of OA has been suggested [27,28,29,30,31,32,33]. It has been found that OA could decrease the growth rate of *Dunaliella tertiolecta* and other non-OA producing algae such as *D. salina, Thalassiosira weissflogii*, *Gambierdiscus toxicus* and *Coolia monotis* [30,31,32]. However, species producing OA were found to be resistant to the deleterious effects of OA [30,31]. This sensitivity of non-OA producing algae to the effect of OA suggests potential functions of OA as a deterrent against settlement of other microalgae adjacent to *P. lima* cells or an anti-grazing chemical to prevent other organisms grazing [31,33]. However, there is not sufficient evidence supporting these functions of OA, and the biological function of OA remains elusive.

As described above, *ABCB*, *ABCC* and *ABCG* are involved in many physiological and biochemical processes in various organisms, and play an important role in transmembrane transport of secondary metabolites and exogenous substances, and anti-autotoxicity [34,35]. Some studies have shown that OA in *P. lima* was compartmentalized in chloroplasts or vacuoles, which could prevent OA from affecting the activity of protein phosphatase in *P. lima* cells [36,37,38]. Based on the anti-autotoxicity mechanism of ABC transporters in plant cells, it is speculated that ABC transporters may transport toxic secondary metabolites such as OA to some organelles to protect itself from harmful effects.

From the above, we speculate that ABC transporters might be implicated in transporting or sequestrating endogenous secondary metabolites such as OA and xenobiotic pollutants like heavy metals, and that OA may function as an anti-grazing chemical to prevent other grazing organisms. The aims of this current study are: (1) to provide information on the character of ABC transporter genes in dinoflagellate and their potential functions in adaption to the surrounding environment in *P. lima*; (2) to give some evidence for OA as an anti-grazing chemical; and (3) to analyze the potential role of ABC transporters in OA transportation in *P. lima*. For these, complete cDNA sequences of *ABCB1*, *ABCC1* and *ABCG2* in *P. lima* were firstly cloned by RACE, and subsequently, their expression patterns were characterized under different conditions, especially under heavy metal stress. Secondly, *P. lima* was observed to produce OA production in the presence of *Artemia salina*, a suitable model species to assess the toxicity of marine benthic dinoflagellates [39]. Finally, given that algal toxin production is governed by multiple intricate physiological and ecological factors, such as the imbalance of nutrient ratio, the availability of nutrients, environmental pollutants, and grazing pressure [40,41,42,43,44,45], the relationship between the expression of three ABC genes and OA production under different conditions (nutrient limitations, different sources of nutrients, heavy metal stress, grazing pressure) was investigated. 

## 2. Results

### 2.1. Sequence Analysis of ABCs Gene

The sequences obtained by RACE and local blast were assembled to full-length transcripts of *ABCB1*, *ABCC1* and *ABCG2* in *P. lima*. The full-length cDNA of *ABCB1*, *ABCC1* and *ABCG2* gene were identified to be 3951, 4259 and 2404 bp, respectively, which have been deposited in NCBI database with the accession numbers (Genbank: MK334304, MK334306 and MK334305). Lengths of open reading frame (ORF) of *ABCB1*, *ABCC1* and *ABCG2* cDNA were found to be 3834, 4098 and 1878 bp in *P. lima* as deduced by ORF finder (http://www.ncbi.nlm.nih.gov/gorf/gorf.html). Correspondingly, the sizes of predicted protein were 138.3 kDa (1277 aa), 147.9 kDa (1365 aa) and 69 kDa (625 aa), and the values of theoretical pI were 5.77, 6.02 and 9.13, respectively. In total, the 5′- and 3′- untranslated regions (UTRs) of *ABCB1*, *ABCC1* and *ABCG2* were 60 and 57 bp, 93 and 68 bp, 140 and 386 bp in *P. lima*, respectively. 

The amino acid sequence analyses showed that the ABCB1 and ABCC1 in *P. lima* had all typical structural elements of ABC transporters with two TMDs subunits and two NBDs, while the ABCG2 had only one TMD and one NBD. However, all the NBDs contained typical and highly conserved motifs of ABC transporters including Walker A, Walker B, ABC signatures and A loops upstream of the Walker A regions (Figure 1). Polyphobius algorithm analyses demonstrated that twelve putative transmembrane helices existed in the ABCB1 transporters, associated with two TMDs with each containing six transmembrane helices, but there were nine and six putative transmembrane helices in ABCC1 transporter and ABCG2 transporter, respectively (Figure 1). Subcellular localization analyses predicted that the three ABC transporters were all located on the cytoplasmic membrane by using CELLO [46], targetP v1.1 [47], ProtComp 9.0 and Euk-mPLoc 2.0 [48] (Appendix A).

Amino acid sequences of ABCB1, ABCC1 and ABCG2 in *P. lima* matched closely to ABCB1, ABCC1 and ABCG2 proteins from various organisms by NCBI BLASTp (Appendix A). Phylogenetic analyses of ABCB1/MDR1 transporters from different organisms revealed that the ABCB1/MDR1 proteins from *P. lima* and *Symbiodinium microadriaticum* showed the closest relationship, forming an independent branch, then clustering into the other branch containing *Fistulifera solaris* (Appendix A). Similarly, ABCC1 proteins from *P. lima* had the closest relationship to *S. microadriaticum* and *Tetraselmis* sp. GSL018 (Appendix A); ABCG2 proteins from *P. lima* shared the closest relationship with *Tetraselmis* sp. GSL018, *Bathycoccus prasinos* and *Cyanidioschyzon merolae* (strain 10D) (Appendix A). 

### 2.2. Content of OA under Different Conditions

Tributyltin (TBT) is the most important source of organotin compounds in the marine environment, which has been extensively used as a biocide in anti-fouling paint, commonly known as bottom paint. Studies have shown its deleterious effect on many levels of the ecosystem, including invertebrates and vertebrates, even humans [49,50]. As demonstrated in Figure 2, *P. lima* exhibited differential growth rate under different conditions. N-limitation, P-limitation, Cu^2+^ and TBT negatively impacted the growth of *P. lima*. *P. lima* grew well when NaH_2_PO_4_, glycerophosphate or ATP were provided as phosphorus sources, and NaNO_3_ and urea used as nitrogen sources. However, when NH_4_Cl was employed as a nitrogen source, the density of *P. lima* showed a downward trend, suggesting that the growth of *P. lima* was significantly depressed.

OA production varied in different growth phase. The *P. lima* cells accumulated the most OA during plateau phase (on the 50th day), followed by late logarithmic growth phase (on the 38th day), and exponential phase (on the 20th day). N-limitation, P-limitation, and Cu^2+^ in 1575 nM and 5039 nM were found to induce OA production in *P. lima*, while TBT exposure (2.7 nM, 8.5 nM) decreased OA production (Figure 3A,E,F).

Under different sources of nitrogen, OA production in the NH_4_Cl group was significantly lower than NaNO_3_ (*p* < 0.05), but there was no significant difference in OA content between urea group and NH_4_Cl group, as well as between NaNO_3_ group and urea group (Figure 3C). When NaH_2_PO_4_ was used as the phosphorus source, OA production in *P. lima* was significantly reduced with respect to glycerophosphate or ATP as phosphorus sources (Figure 3D). 

In the presence of *A. salina*, the content of OA in *P. lima* was significantly higher than that of control (*p* < 0.05) (Figure 3B). After 48 h, the survival rate of *A. salina* exposed to *P. lima* were 72.8 ± 3.6%, which was distinctly lower than their control counterparts (91.1 ± 2.5%), suggesting the toxicity of *P. lima* to the *A. salina*. 

### 2.3. Transcriptional Responses of ABCB1, ABCC1 and ABCG2 in P. lima under Different Conditions

As shown in Figure 4, the expression of *ABCB1*, *ABCC1* and *ABCG2* was greatly regulated by specific environmental factors. Under nitrogen or phosphorus-limitation, the expressions of *ABCB1* and *ABCG2* in *P. lima* were up-regulated, whereas *ABCC1* was significantly reduced (*p* < 0.05) at the 38th day under N-limited condition (Figure 4A–C). 

The *P. lima* cells exhibited different expression profiles of *ABCB1*, *ABCC1* and *ABCG2* when cultured under different sources of nitrogen and phosphorus. The expression levels of *ABCB1* and *ABCC1* were more sensitive to different nitrogen and phosphorus substrates compared to that of *ABCG2*. *ABCB1* transcript abundance in *P. lima* was lower when urea and NaH_2_PO_4_ as nitrogen and phosphorus source compared with other nutrient sources assayed (Figure 4D,E). Phosphorus sources displayed some effects on the expression of *ABCC1* mRNA in *P. lima*. *ABCC1* performed high expression in *P. lima* when glycerophosphate as phosphorus source, followed by ATP, finally NaH_2_PO_4_ (Figure 4E). As for the *ABCG2*, it had almost the same expression level in *P. lima* under different nutrient sources assayed in our study (Figure 4D,E).

Cu^2+^ in some concentrations distinctly up-regulated the expression of *ABCB1* (1575 nM, 5039 nM), *ABCG2* (1575 nM, 5039 nM) and *ABCC1* (5039 nM) in *P. lima* (Figure 4G–I). Similarly, TBT exposure could also induce the expression of *ABCB1*. Under 2.7 nM and 8.5 nM of TBT, the expression of *ABCB1* was significantly higher than that of the control (Figure 4J).

In addition, the expressions of *ABCB1* and *ABCG2* were significantly up-regulated in the presence of *A. salina* (Figure 4F). 

## 3. Discussion

*ABCB1*, *ABCC1* and *ABCG2*, i.e., the members of the ABC transporter family in *P. lima* obtained in this study, have typical characteristics of their respective sub-families. Consistent with features of ABCB family, ABCB1 protein in *P. lima* is a full transporter arranged as “TMD-NBD-TMD-NBD", with 12 transmembrane helices and approximately half of α-helix. It had highly conserved motifs “Walker A" and “Walker B", and “ABC signature", sharing high homology with other species ABCB1/MDR1 protein. ABCG proteins not only have a full-structured “NBD-TMD-NBD-TMD" molecule, but also have a half-structured “NBD-TMD" molecule [51]. The ABCG2 protein in *P. lima* is a half-transporter with six transmembrane helices at C-terminus. 

ABCC subfamily proteins are full-transporters with diverse functions, such as ion transport and translocation of various xenobiotics and endogenous compounds. As one of long MRPs, ABCC1 has an additional fifth domain, named MSD0 at N terminus in human [52]. However, ABCC1 protein in *P. lima* only have two TMDs, with each spanning domain followed by a NBD. Similarly, the dinoflagellate *S. microadriaticum* was proposed to have only 6 transmembrane helices, but *Tetraselmis* sp. GSL018 have 12 transmembrane helices. The difference in number of transmembrane helices somewhat indicates the complexity of ABC family in microalgae. 

All the ABCBs in *Arabidopsis* are localized to the plasma membrane. In general, plant ABCB1 are associated with auxin transport, and metal stress tolerance [14]. Similarly, most ABCCs in plants are featured as vacuolar localized proteins, and few have been shown to reside on the plasma membrane [53]. ABCC1 has been demonstrated to be involved in anthocyanin accumulation in vacuoles, and transport of glutathione-conjugates and folate [5]. Half-size ABCGs in plants are localized in the plasma membrane, mitochondrial membrane, chloroplast membrane and cytoplasm. It has been found that ABCG2 is required for synthesis of an effective suberin barrier in roots and seed coats [54]. Based on the sequences obtained, our in silico analyses predicted these three ABC transporters could be localized to the cytoplasmic membrane.

Some studies have shown that OA production in *P. lima* was increased under nitrogen and phosphorus limitation [25,27,40], and its production was highly varied dependent on the sources of nitrogen and phosphorus [27,28]. Consistently, we found that nitrogen and phosphorus limitation could significantly promote the OA production in *P. lima*. OA production was varied under different nitrogen and phosphorus sources. When NH_4_Cl was used as a nitrogen source, the growth rate of *P. lima* and OA production were significantly decreased compared with other N substrates, indicating the lethal effect of ammonium at high concentration on *P. lima*, though it has been proposed to prefer ammonium over other N substrates in cultures [27].

Heavy metals are usually regarded as a main anthropogenic contaminant in coastal and marine environments all over the world [55], which can also affect the growth and toxin production of dinoflagellate [41,42,43]. However, not all metal elements could promote the production of microalgae toxins. For example, selenium at pM level could induce yessotoxin production in *Protoceratium reticulatum*, but iron and cobalt did not [42]. Tributyltin has been extensively used as a biocide in anti-fouling paint, fungicide in agricultural activities, and industrial catalyst in the production of polyurethane foams, etc. As the most important source of organotin compounds in the marine environment, TBT is doomed to be highly persistent due to its reversible adsorption to sediments [50]. Owing to the wide use of Cu in industrial activities and other fields (as an important ingredient in many algaecides and herbicides), Cu accumulation in marine sediments has also been becoming a great concern in marine ecosystems [43]. Here, we found that OA production in *P. lima* was significantly increased under higher concentrations of Cu^2+^ (1575 nM and 5039 nM), but decreased under higher concentrations of TBT (2.7 nM and 8.5 nM). Copper is one of the most common pollutants in the marine environment, therefore, it should be comprehensively considered when assessing the harmful effects of benthic dinoflagellates.

It has been found that some phycotoxins can function as inducible defensive chemicals against grazing or reducing the number of grazers to ensure that they were not cleared by grazers [44,56,57]. Nowadays, numerous studies have shown that this inducible reaction is common in phytoplankton, which can reduce the grazing pressure to maintain the survival of the population [57,58]. Selander et al. found that waterborne cues from grazing copepods could increase the cell-specific paralytic shellfish poisoning (PSP) toxin [45]. Dang et al. reported that some grazers could increase the toxicity of *K. mikimotoi* [44]. Here, we found that OA production in *P. lima* was significantly increased in the presence of *A. salina*, suggesting the possible anti-grazing role of OA. During the experiment, the survival rate of *A. salina* exposed to *P. lima* was distinctly lower than their control counterparts, which is in line with a previous result on the harmful impact of *P. lima* on behavior and survival of *A. salina* [39]. The promotion of grazing pressure on OA production may further increase the accumulation of toxins in the food chain, and increase the risk of toxins.

According to our predictions, the expression levels of ABC transporters varied under different environmental conditions. Nitrogen and phosphorus deficiency induced the expression of *ABCB1* and *ABCG2* in *P. lima*. Nitrogen and phosphorus substances, metal ions, and even grazers were shown to regulate the expression of ABC transporters. However, there was no significant correlation observed between the expression of *ABCB1*, *ABCC1* and *ABCG2* genes and OA production. ABC transporters are important carriers for the discharge and compartmentalization of toxic substances and external pollutants, playing an important role in the resistance of algae to secondary metabolites and environmental pollutants [59]. It has been proposed that OA might be synthesized in chloroplast, and stored in chloroplast or vacuoles around the cytoplasm [37]. Furthermore, OA has been confirmed to be eliminated by P-glycoprotein over the apical membrane [60]. The subcellular localization of the three ABC transporters and the absence of exact correlation between their expressions and OA production seem to diminish the possibility of three transporters transporting OA in *P. lima.* Nevertheless, DSP toxins-producing dinoflagellate has been shown to secrete toxins into water [61], which suggests the underlying role of the three ABC transporters in transporting DSP toxins from the cells. The production and transportation of DSP toxins in *P. lima* are very complicated, and more studies are warranted. 

Previous studies have shown that ABC transporters might involve in the detoxification of heavy metals in plants. Gaillard et al. found that *AtMRP6* expression was up-regulated in *Arabidopsis thaliana* under cadmium stress [9]. Lee et al. observed the up-regulation of *AtABCG40/AtPDR12* in the shoots of wild plants after exposed to Pd^2+^, which in turn conferred Pd resistance in *Arabidopsis* [62]. Kim et al. demonstrated that AtPDR8 localized in the plasma membrane of epidermal cells which mediated Cd^2+^ efflux by measuring the ^109^Cd radioactivity in *Arabidopsis* [10]. In this study, we found that *ABCB1*, *ABCC1* and *ABCG2* transcripts in *P. lima* were up-regulated at higher Cu^2+^ concentration, suggesting the potential role of ABC transporters in detoxification of heavy metals in *P. lima*. However, the exact role of ABC transporters in dinoflagellate against metal ion in surrounding waters remains to be investigated. The three ABC transporters in *P. lima* might be regulated by various interrelated metabolic changes as in plant [63]. 

In conclusion, our results demonstrated intriguing data about OA production and expression patterns of various ABC transporter genes. Cu^2+^ at some concentrations induced OA production, whereas TBT inhibited OA accumulation. The grazer *A. salina* could induce OA production, and *P. lima* displayed some toxicity to the grazer, indicating the possible anti-grazing property of OA. Under N- and P-limited conditions, the expression levels of *ABCB1* and *ABCG2* in *P. lima* were significantly up-regulated. Nitrogen and phosphorus substrates, metal ions, and even grazer could also regulate the expression of ABC transporters. However, we could not find any significant correlation between OA production and the expression of the three ABC transporters in *P. lima*, implying the complexity of OA transportation in *P. lima*. Higher concentration of Cu^2+^ could up-regulate *ABCB1*, *ABCG2* and *ABCC1* transcripts in *P. lima*, suggesting the potential role of ABC transporters in dinoflagellate against metal ion in surrounding waters.

## 4. Materials and Methods

### 4.1. Algal Culture 

*P. lima* (CCMP 2579) was kindly provided by National Center for Marine Algae and Microbiota (NCMA, formerly CCMP). The strain was grown as batch cultures in sterile Erlenmeyer flasks containing f/2 medium, which was filter-sterilized through 0.22-μm filters (Jinjing, China). The cultures were grown at 20 ± 1 °C in an artificial climate incubator (Jiangnan Instrument Factory, Ningbo, China), where cool-white fluorescent tubes provided an irradiance of 58 μmol photons m^−2^ s^−1^ with a 12/12 h light/dark cycle.

### 4.2. Cloning of ABC Transporter Genes

Algal cells were collected by centrifugation (4500× *g*, 4 °C, 2.5 min) from 50 mL of *P. lima* culture (1.2 × 10^7^ cells/L) after 38 days, then ground to powder in a mortar with liquid nitrogen. Total RNA was extracted with a Hipure Plant RNA Mini kit (Magen, Guangzhou, China) according to the manufacturer’s instruction, and retro-transcribed to cDNA using a PrimeScript™ II 1st Strand cDNA Synthesis kit (TaKaRa, Dalian, China). 

Initial cDNA sequences were obtained by comparing ABC transporter sequences of similar species such as *ABCB1* and *ABCC1* of *Cyanidioschyzon merolae* strain 10D and *ABCG2* of *Tetraselmis* sp. GSL018 in GenBank with the Trinity.fasta transcriptome (NCBI database: SRR5,768,053, SRR5,796,841, SRR5,796,842 and SRR5,796,843) of P. lima obtained in our laboratory. Based on the partial sequences obtained, 5′-Full RACE was performed to generate full-length cDNA sequences of *ABCB1*, *ABCC1* and *ABCG2*. Related primers for 5′-RACE are listed in Table 1. Touch down PCR was performed and related PCR profiles were as follows: 94 °C for 3 min, 30 cycles of 94 °C (30 s), 65 °C (30 s), 72 °C (2 min), followed by 15 cycles of 94 °C (30 s), 50 °C (30 s) and 72 °C (2 min), finally 72 °C for 10 min. Same procedures were performed for *ABCC1* and *ABCG2*. Fragments were purified by using a Gel & PCR Clean Up kit (Omega, Guangzhou, China), then cloned into the vectors pMD18-T (TaKaRa, Dalian, China) according to the manufacturer’s instructions, and sequenced. 

The sequences were assembled and analyzed by using tools at the Expert Protein Analysis System Server (http://expasy.org/) and NCBI BLAST search engines. TMHMM (http://www.cbs. dtu.dk/services/TMHMM/) was employed to identify transmembrane helices within the amino acid sequences of ABCB1, ABCC1 and ABCG2. Subcellular localization of the three ABC transporters were predicted using CELLO (Xinzhu, Taiwan, China [46], targetP v1.1 [47], ProtComp 9.0 (http://www.softberry.com/berry.phtml?topic=protcomppl&group=programs&subgroup=proloc) and Euk-mPLoc 2.0 [48]. Homology modeling was carried out in term of the amino acid sequences of ABCB1, ABCC1 and ABCG2 to predict the tertiary structure by using the Swiss-model program (http://swissmodel.expasy.org/). InterPro (http://www.ebi.ac.uk/interpro/) was employed to predict domains of the ABCB1, ABCC1 and ABCG2 amino acid sequences. Transmembrane protein models were mapped using Adobe Photoshop CC 2015 software (https://www.adobe.com/cn/). Multiple sequence alignments and determinations of similarity among amino acid sequences of ABCB1, ABCC1 and ABCG2 transporters from different organisms were carried out by using Clustal X2 (http://www.clustal.org/clustal2/) and BLAST on NCBI. Phylogenetic trees of *ABCB1*, *ABCC1* and *ABCG2* transporter sequences from different organisms were constructed by Neighbor-joining (NJ) method using MEGA version 7.0 (Philadelphia, PA, USA).

### 4.3. Detection of OA in P. lima

Algal cells were collected by centrifugation (4500× *g*, 4 °C, 2.5 min) from 50 mL of *P. lima* culture, then ground to powder in a mortar with liquid nitrogen. Sequentially, 2 mL of 80% methanol was added to the cells and pelleted twice to extract toxins. The supernatant was collected by centrifugation (3000× *g* for 10 min) and diluted to 10 mL with 80% methanol for OA detection. OA was measured by using an Okadaic Acid (DSP) ELISA test kit (Abraxis, Warminster, PA, USA) according to the manufacturer’s instruction.

### 4.4. Reverse Transcription-Quantitative PCR (RT-qPCR)

Expression of *ABCB1*, *ABCC1* and *ABCG2* mRNA in *P. lima* was evaluated by RT-qPCR. Total RNA was extracted by using a Hipure Plant RNA Mini kit (Magen, Guangzhou, China) according to the manufacturer’s instruction. RNA integrity and genome DNA contamination were tested by agarose gel electrophoresis. The concentration and OD260/280 ratio of RNA were evaluated by NanoDrop^®^ND-1000 (NanoDrop, Waltham, MA, USA). First strand cDNA for each sample was generated from 1 μg of total RNA using a HiScript Q RT SuperMix for qPCR (Vazyme, Nanjing, China). Specific primers were designed by using Primer Premier 6.0 (http://www.premierbiosoft.com) (Table 2). The software geNorm (https://genorm.cmgg.be), NormFinder (https://www.moma.dk) and Bestkeeper (https://www.gene-quantification.de/bestkeeper.html) were employed to screen internal genes to normalize expressions of ABCs from the five common housekeeping genes such as 18S RNA, β-actin, β-tubulin, GAPDH and Calm. β-actin and β-tubulin were chosen as internal genes to normalize expression of target genes due to their most stable expressions.

RT-qPCR was performed in a Bio-Rad CFX96 Real Time PCR System (Bio-Rad, Hercules, CA, USA) with AceQ qPCR SYBR Green Master Mix (Vazyme, Nanjing, China) according to the manufacturers’ instructions. PCR profiles were as follows: 95 °C for 5 min; 40 cycles of 95 °C (30 s), 60 °C (30 s); 94 °C for 15 s, 60 °C for 30 s and 95 °C for 15 s. 

Relative mRNA expression of ABCs gene in different samples was calculated by formula NRQ [64] with Bio-Rad CFX Manager 3.0 (http://www.advanceduninstaller.com/Bio-Rad-CFX-Manager-3_0-9f36ee671f747e20a29b7061a3edb5b1-application.htm), in which inter-run calibration algorithms were considered. A standard curve was generated to determine the efficiency of PCR amplification. Amplification efficiency for each reaction was ranged from 0.958 to 1.146, while the correlation coefficient was ranged from 0.928 to 0.999. The differences in amplification efficiency between target genes and internal genes were less than 10%.

### 4.5. Expression of ABC Transports in P. lima under Different Conditions

To learn responses of the three ABC transporter genes to different conditions and to gain information about the relationship between ABC transporters expression and OA production, four experiments were performed: culture under nutrient-limited conditions (N-limited, P-limited), culture under different sources of nutrients (N or P), exposure to different heavy metal, and exposure to *A. salina*. All treatments were performed in triplicate.
*P. lima* under nutrient-limited conditions. *P. lima* in an initial cell density of 1.2 × 10^7^ cells/L was grown under N-limited conditions (with 17.7 μM NO_3_^−^), P-limited condition (1.81 μM PO_4_^3−^) or nutrient-sufficient conditions (with 883.0 μM NO_3_^−^, 36.3 μM PO_4_^3−^). Trace metals and EDTA were at the levels corresponding to f/2 medium in all cultures. Algal cells were collected on days 20, 38, and 50 for OA extraction, RT-qPCR analysis and morphologic observation, etc.*P. lima* under different nutrients sources (nitrogen or phosphorus) conditions. *P. lima* in an initial cell density of 1.2 × 10^7^ cells/L was grown under 883 μM of NaNO_3_, urea or NH_4_Cl, where other nutrients were the same as the f/2 medium. Algal cells were collected on the 38^th^ day for OA detection and RT-qPCR analysis. Similar experiments were performed by employing NaH_2_PO_4_, glycerophosphate or ATP (36.3 μM) as sources of phosphorus.*P. lima* incubated in f/2 medium with Cu^2+^ or TBT. *P. lima* in an initial cell density of 1.2 × 10^7^ cells/L was incubated in f/2 medium with Cu^2+^ (504 nM, 1575 nM and 5039 nM) or TBT (0.85 nM, 2.7 nM and 8.5 nM) as described Couet et al. [43]. As for the control, f/2 medium was employed. After 7 days of culture, algal cells were collected for OA detection and RT-qPCR analysis.*P. lima* exposure to *A. salina*. *P. lima* in an initial cell density of 1.2 × 10^7^ cells/L was cultured in f/2 medium with or without 180 of *A. salina* as described by Dang et al. [44]. The *A. salina* individuals were hatched from the spawns, which were purchased from Ocean University of China in Qingdao, China. The animals were cultured in 2000 mL of seawater sterilized with a salinity of 27 at 20–25 °C for 7 days, and fed with the green alga *Tetraselmis subcordiformis.*

Some viable animals (8.0–10.0 mm in size) were transferred into fresh f/2 medium and starved for 72 h before the experiment. During the experiment, dead *A. salina* was removed every 24 h and replaced with healthy and lively *A. salina* individuals. After 48 h, microalgal cells were collected for OA detection and RT-qPCR analysis.

### 4.6. Statistical Analyses

Statistical analyses were carried out by using the software SPSS 19 (Armonk, NY, USA). All data were expressed as mean values ± standard deviation. Student’s t-test with a 95% confidence interval was used to compare differences between control and treatment groups except for different nutrient sources treatments. LSD t-test was employed to compare differences between means under various nitrogen or phosphorus sources.

## Figures and Tables

**Figure 1 marinedrugs-17-00259-f001:**
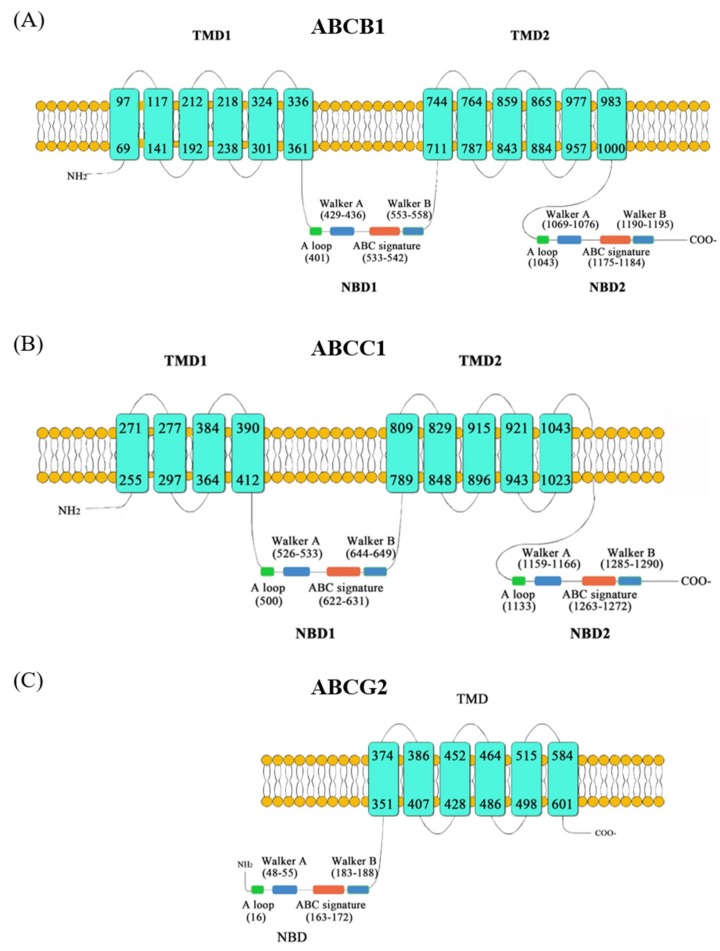
Topologies of ABCB1 (**A**), ABCC1 (**B**) and ABCG2 (**C**) protein in *P. lima* with transmembrane domains (TMDs) as predicted by the Polyphobius algorithm and nucleotide binding domains (NBDs) indicated by A loop, Walker A and B, and the ABC signature motifs. (**A**) ABCB1 has two TMDs with 12 transmembrane helices, two NBDs with A loop, Walker A and B, and the ABC signature motifs; (**B**) ABCC1 has two TMDs with 9 transmembrane helices, two NBDs with A loop, Walker A and B, and the ABC signature motifs; (**C**) ABCG 2 has one TMD with 6 transmembrane helices, one NBD with A loop, Walker A and B, and the ABC signature motifs.

**Figure 2 marinedrugs-17-00259-f002:**
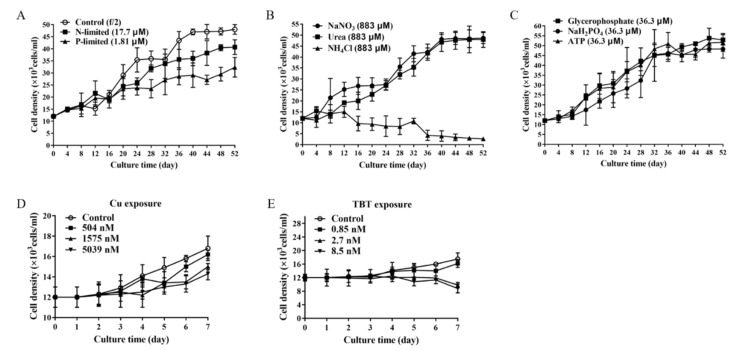
The growth curve of *P. lima* under different environmental conditions. (**A**) nutrient limitations (N-limited, P-limited); (**B**) different sources of nitrogen (NaNO_3_, urea and NH_4_Cl); (**C**) different sources of phosphorus (NaH_2_PO_4_, glycerophosphate and ATP); (**D**) Cu exposure (504 nM, 1575 nM and 5039 nM); (**E**) TBT exposure (0.85 nM, 2.7 nM and 8.5 nM). The values are expressed as mean ± SD (n = 3).

**Figure 3 marinedrugs-17-00259-f003:**
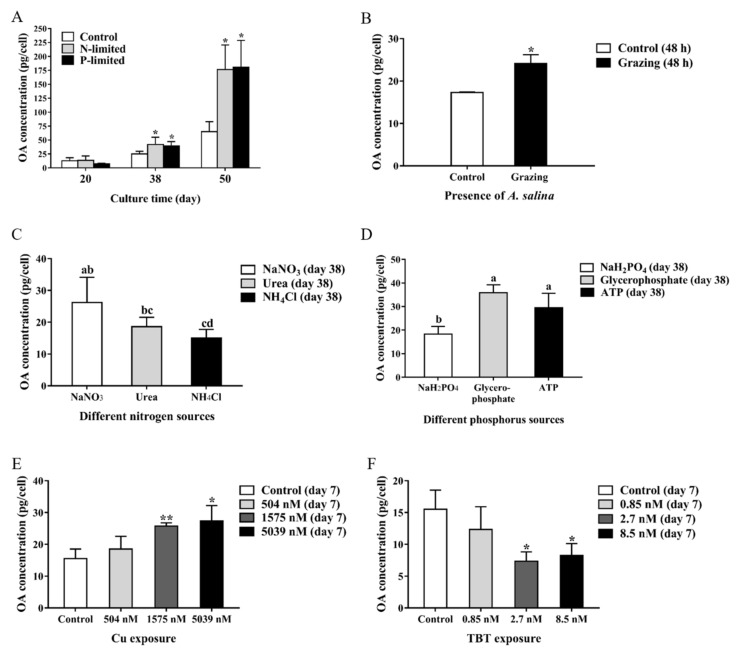
Okadaic acid content (pg cell^−1^) in *P. lima* under different conditions. (**A**) nutrient limitations; (**B**) exposure to *A. salina*; (**C**) different nitrogen sources; (**D**) different phosphorus sources; (**E**) Cu exposure; **F**, TBT exposure. The values are expressed as mean ± SD (n = 3). In A, B, E and F, asterisks indicate statistically significant differences between control and treatment groups (t-test, * *p* < 0.05; ** *p* < 0.01). In C and D, bars of respective treatment followed by the same letter are not significantly different at *p* < 0.05 (LSD t-test). In C, OA production in NH_4_Cl group (indicated with cd) was significantly lower than NaNO_3_ (indicated with ab) as nitrogen substrate ( *p* < 0.05), but there was no significant difference between urea group (indicated with bc) and NH_4_Cl group (indicated with cd), as well as between NaNO_3_ group (indicated with ab) and urea group (indicated with bc). In D, OA content in NaH_2_PO_4_ group (indicated with b) was significantly lower than glycerophosphate (indicated with a) or ATP (indicated with a) as phosphorus sources, but there was no significant difference between glycerophosphate group (indicated with a) and ATP group (indicated with a).

**Figure 4 marinedrugs-17-00259-f004:**
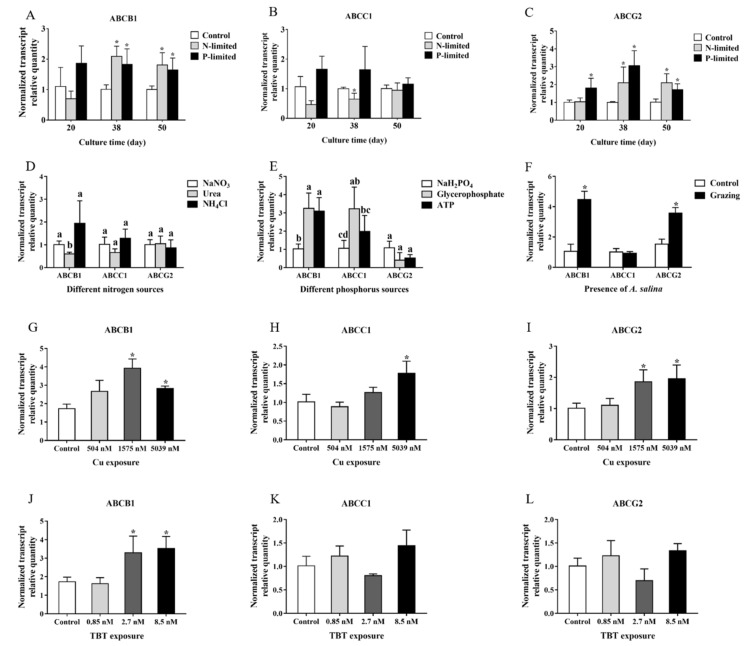
Expression levels of *ABCB1*, *ABCC1* and *ABCG2* mRNA in *P. lima* under different conditions. (**A**) *ABCB1* under nutrient limitations; (**B**) *ABCC1* under nutrient limitations; (**C**) *ABCG2* under nutrient limitations; (**D**) *ABCB1*, *ABCC1* and *ABCG2* under different nitrogen sources (day 38; (**E**) *ABCB1*, *ABCC1* and *ABCG2* under different phosphorus sources (day 38); (**F**) *ABCB1*, *ABCC1* and *ABCG2* in the presence of *A. salina* (48 h); (**G**) *ABCB1* under Cu exposure (day 7); (**H**) *ABCC1* under Cu exposure (day 7); (**I**) *ABCG2* under Cu exposure (day 7); (**J**) *ABCB1* under TBT exposure (day 7); (**K**) *ABCC1* under TBT exposure (day 7); (**L**) *ABCG2* under TBT exposure (day 7). The values are expressed as mean ± SD (n = 3). In A, B, C, F, G, H, I, J, K and L, asterisks indicate statistically significant differences between control and treatment groups (t-test, * *p* < 0.05). In D and E, bars of respective treatment followed by the same letter are not significantly different at *p* < 0.05 (LSD t-test). In D, *ABCB1* transcript abundance when urea as nitrogen (indicated with b) was lower than NaNO_3_ group (indicated with a) and NH_4_Cl group (indicated with a) (*p* < 0.05). There was no significant difference in ABCC1 and ABCG2 expression between different groups (all indicated with a). In E, ABCB1 transcript abundance when NaH_2_PO_4_ as phosphorus (indicated with b) was lower than ATP group (indicated with a) and glycerophosphate group (indicated with a) (*p* < 0.05). ABCC1 expression in NaH_2_PO_4_ group (indicated with cd) was significantly lower than glycerophosphate (indicated with ab) as phosphorus sources, but there was no significant difference between glycerophosphate group (indicated with ab) and ATP group (indicated with bc), as well as between NaH_2_PO_4_ group (indicated with cd) and ATP group (indicated with bc). As for ABCG2, there was no significant difference (all indicated with a).

**Table 1 marinedrugs-17-00259-t001:** Primers for 5′-RACE.

Primer Name	Primer Sequence (5′–3′)	Target
M13-47	CGCCAGGGTTTTCCCAGTCACGAC	5′-RACE
RV-M	GAGCGGATAACAATTTCACACAGG	5′-RACE
AAP	CGCGTCGACTAGTACGGGGGGGGGG	5′-RACE
AUAP	CGCGTCGACTAGTAC	5′-RACE
ABCB1-5GSP1	GCAGAACCGCACGACGACCT	5′-RACE
ABCB1-5GSP2	GCTCAGACCATGCGCCACAG	5′-RACE
ABCC1-5GSP1	CTGTCATACAGGGGTTGTGGCTCGCTG	5′-RACE
ABCC1-5GSP2	GCTCCAGCTTGCCGGTATCAG	5′-RACE
ABCG2-5GSP1	GCATCCTCTTGTTCCACATACGC	5′-RACE
ABCG2-5GSP2	GACAGTGTTCGCATAGAAGGTG	5′-RACE

**Table 2 marinedrugs-17-00259-t002:** Specific primers for RT-qPCR used in this paper.

Gene Name	Primer Sequence (5′–3′)	Length (bp)
ABCB1	F: ACGTCGGTGAAGGACAACATC	128
	R: CCGACGAAGGTGTTGAACTTC	
ABCC1	F: ATGTTGACGGCAACGCAGCTC	121
	R: AGAAGCTGGAACGTCCACGTG	
ABCG2	F: AGCATGGTCCGATTGCCATG	123
	R: TTAGCTGCCCTGGATCACAC	
β-tubulin	F: GTTGCCTCGTTGTAGTAGACG	106
	R: TTTGGGAGGTGATTTCCGACG	
β-actin	F: GCCGACTTCATCTCTGTGTC	101
	R: GGCTACTCCTTCACCACTACG	
18S	F: CCGACTTAGCAGAAGGGTTG	104
	R: CAGCAGACGCCATACGACTA	
Calm	F: GCCATGAGGGACAGGAACT	131
	R: AAGGAGCTTGGAACCGTGAT	
GAPDH	F: CCCACTCGTTGTCGTACCAG	105
	R: CGGATTTCGTGAGCAACAA	

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
