# Peer review of "ABC Transporters in Prorocentrum lima and Their Expression Under Different Environmental Conditions Including Okadaic Acid Production"

_marinedrugs, 2019, doi:10.3390/md17050259_

Reviewer 1 Report

The work by Gu and colleagues describes the identification of three genetic sequences putatively encoding distinct ABC transporters in P. lima and investigates their expression in conditions that are associated with the production of okadaic acid in this organism, simulating nutrient and grazing stress. The manuscript is well written, the results are generally well presented. Although the main outcomes do not add much new knowledge on the topic, I found that the results have been properly discussed and put well in context with the existing research. I, however, have several concerns that will need to be addressed prior recommending the acceptance of this manuscript.

-       Line 124 – 125. The authors state that the P.lima ABC sequences closely match those of other organisms. To support this statement, BLAST results and closest BLAST hits will need to be included, ideally in the Supplementary Information

-       In general, although the topic has been appropriately reviewed and contextualized in the introduction, the author did not provide a clear formulation of their hypotheses on which the experiments were designed. As such, it is not immediate to understand the rationale behind the specific experiments, to evaluate the experimental design and the interpretation/discussion of the results. I invite the authors to add this, as it is a central part of a scientific investigation.

-       For example, it is not clear why these specific N and P sources were chosen in the described experiments, in particular it is not clear to me why ATP has been used as alternative phosphorous source.

-       Line 134 – Although explained later, TBT acronym will need to be explained in full the first time it’s mentioned

-       Figure 2. Although reported in the methods, it would be helpful to report the concentration of N and P sources on the graphs, as done for Cu and TBT (it applies to other figures too), as well as the incubation time for cultures subjected to Cu and TBT treatments.

-       Figure 3/4. At what time point/ growth stage were these analyses carried out?

-       Figure 4. It is not clear what calculation method and procedures the authors have used to obtain the gene expression values. Although some references have been provided, some description on method and statistics would help understand the results. Also, I find incorrect the labelling of the y axes “mRNA expression level”. Depending on the quantification method used, this will need to be changed accordingly. I assume this might be “normalized transcript relative quantity”.

-       On line 265 it is stated that “ OA has been confirmed as a substrate for ABC transporters [54]”. However, the cited reference does not report any conclusive result on this, but only indirect evidence of interaction, hypothesizing substrate or inhibitor relationship.

-       Relevant literature [55] supporting the (not formulated) hypothesis that ABC transporters are regulated at the transcriptional level in P.lima is only cited and described on line 273, while it would have been more helpful in understanding the experimental design, if cited earlier.

-       Line 299 – were the P. lima cultures axenic monocultures? The presence or absence of bacteria in the experiments might have influenced the results

-       Line 341 Three biological replicates were used for gene expression analyses. It is not clear though if technical replicates were included too.

Author Response

Line 124 – 125. The authors state that the P. lima ABC sequences closely match those of other organisms. To support this statement, BLAST results and closest BLAST hits will need to be included, ideally in the Supplementary Information

Answer: Thank you very much indeed for your suggestions. The related expression has been added in the revised manuscript, and BLAST results has been added in Supplementary Information.

In general, although the topic has been appropriately reviewed and contextualized in the introduction, the author did not provide a clear formulation of their hypotheses on which the experiments were designed. As such, it is not immediate to understand the rationale behind the specific experiments, to evaluate the experimental design and the interpretation/discussion of the results. I invite the authors to add this, as it is a central part of a scientific investigation.

Answer: Thank you very much indeed for your comments. The related expression has been added in the revised manuscript.

For example, it is not clear why these specific N and P sources were chosen in the described experiments, in particular it is not clear to me why ATP has been used as alternative phosphorous source.

Answer: Thank you very much indeed for your questions. NaNO3, urea, NH4Cl, NaH2PO4 and glycerophosphate have been extensively used as N sources or P sources to learn the capacity of different species of harmful algae (diatoms, dinoflagellates, raphidophytes) to utilize different nutrient sources (Cucchiari et al., 2008; Wang et al., 2011). ATP, an important form of dissolved organic phosphorus (DOP), occurs stably in seawater in significant concentration. ATP utilisation as the P source has been documented in eukaryotic phytoplankton as well as bacteria (Luo et al., 2017). For examples, Chattonella ovata has been found not to utilize organic nitrogen (urea and uric acid), but have ability of using ATP, ADP and inorganic phosphorus compounds (Yamaguchi et al., 2008). Under phosphorus limitation, Prorocentrum donghaiense can utilize various types of phosphoester substrates such as nucleic acids, ATP and lipids as alternative sources of phosphate (Shi et al., 2017). Therefore, we chose the six specific N and P sources in our study.

References

Cucchiari, E.; Guerrini, F.; Penna, A.; Totti, C.; Pistocchi, R. Effect of salinity, temperature, organic and inorganic nutrients on growth of cultured Fibrocapsa japonica (Raphidophyceae) from the northern Adriatic Sea. Harmful Algae 2008, 7, 405–414

Luo, H.; Lin, X.; Li, L.; Lin, L.; Zhang, C.; Lin, S. Transcriptomic and physiological analyses of the dinoflagellate Karenia mikimotoi reveal non-alkaline phosphatase-based molecular machinery of ATP utilisation. Environ. Microbiol. 2017, 19, 4506–4518.

Shi, X.; Lin, X.; Li, L.; Li, M.; Palenik, B.; Lin, S. Transcriptomic and microRNAomic profiling reveals multi-faceted mechanisms to cope with phosphate stress in a dinoflagellate. ISME J. 2017, 11, 2209–2218.

Wang, Z.H.; Liang, Y.; Kang, W. Utilization of dissolved organic phosphorus by different groups of phytoplankton taxa. Harmful Algae 2011, 12, 113-118.

Yamaguchi, H.; Sakamoto, S.; Yamaguchi, M. Nutrition and growth kinetics in nitrogen- and phosphorus-limited cultures of the novel red tide flagellate Chattonella ovata (Raphidophyceae). Harmful Algae 2008, 7, 26–32.

Line 134 – Although explained later, TBT acronym will need to be explained in full the first time it’s mentioned.

Answer: Thank you very much indeed for your suggestions. The related expression has been added in the revised manuscript.

Figure 2. Although reported in the methods, it would be helpful to report the concentration of N and P sources on the graphs, as done for Cu and TBT (it applies to other figures too), as well as the incubation time for cultures subjected to Cu and TBT treatments.

Answer: Thank you very much indeed for your suggestions. The related information has been added in Figure.

Figure 3/4. At what time point/ growth stage were these analyses carried out?

Answer: Thank you very much indeed for your question. The related information has been added in Figures or Figure legend.

Figure 4. It is not clear what calculation method and procedures the authors have used to obtain the gene expression values. Although some references have been provided, some description on method and statistics would help understand the results. Also, I find incorrect the labelling of the y axes “mRNA expression level”. Depending on the quantification method used, this will need to be changed accordingly. I assume this might be “normalized transcript relative quantity”.

Answer: I apologize for the carelessness in description of qPCR. We have replaced “mRNA expression level” by “normalized transcript relative quantity”, and added the related descriptions about calculation method and procedures in the revised manuscript. The software geNorm, NormFinder and Bestkeeper were employed to screen internal genes to normalize expressions of ABCs from five housekeeping genes such as 18S RNA, β-actin, β-tubulin, GAPDH and Calm. β-actin and β-tubulin were chosen as internal genes to normalize expression of target genes owing to their most stable expressions. Relative mRNA expression of ABCs gene in different samples was calculated by formul NRQ (Hellemans et al., 2007) with Bio-Rad CFX Mannager 3.0, in which inter-run calibration algorithms were considered. To determine the efficiency of PCR amplification, a standard curve was generated. Amplification efficiency for each reaction was ranged from 0.958 to 1.146, while correlation coefficient was ranged from 0.928 to 0.999. The differences in amplification efficiency between target genes and internal genes were less than 10%.

References

Hellemans, J.; Mortier, G.; De Paepe, A.; Speleman, F.; Vandesompele, J. qBase relative quantification framework and software for management and automated analysis of real-time quantitative PCR data. Genome Biol. 2007, 8(2), R19.

On line 265 it is stated that “OA has been confirmed as a substrate for ABC transporters [54]”. However, the cited reference does not report any conclusive result on this, but only indirect evidence of interaction, hypothesizing substrate or inhibitor relationship.

Answer: Thank you very much indeed for your comments. This reference did not give direct evidence indeed for that OA is a substrate of ABC transporters. Therefore, this reference has been replaced by another reference, and the related description has been revised. Ehlers et al. (2014) reported that OA at non-cytotoxic concentrations passed the monolayer barrier only to a low degree, and that it was actively eliminated by P-glycoprotein, an ABC transporter, over the apical membrane.

Reference

Ehlers, A.; These, A.; Hessel, S.; Preiss-Weigert, A.; Lampen, A. Active elimination of the marine biotoxin okadaic acid by P-glycoprotein through an in vitro gastrointestinal barrier. Toxicol. Lett. 225, 311-317.

Relevant literature [55] supporting the (not formulated) hypothesis that ABC transporters are regulated at the transcriptional level in P. lima is only cited and described on line 273, while it would have been more helpful in understanding the experimental design, if cited earlier.

Answer: Thank you for your suggestion indeed. We have cited the reference in introduction, and the related description has been added.

Line 299 – were the P. lima cultures axenic monocultures? The presence or absence of bacteria in the experiments might have influenced the results

Answer: In this current study, P. lima was kindly provided by National Center for Marine Algae and Microbiota, USA (NCMA, formerly CCMP). The strain was grown in sterile Erlenmeyer flasks containing f/2 medium. So, P. lima cultures were axenic monocultures. As you commented, bacteria might have indeed influenced the results, including OA production and ABC transporters expression.

Line 341 Three biological replicates were used for gene expression analyses. It is not clear though if technical replicates were included too.

Answer: In our study, the comparative Cq method was used to analyze relative quantity of gene expression as described by Hellemans et al. (2007), in which inter-run calibration algorithms were considered. So technical replicates have not been performed, but three biological replicates were used.

References

Hellemans, J.; Mortier, G.; De Paepe, A.; Speleman, F.; Vandesompele, J. qBase relative quantification framework and software for management and automated analysis of real-time quantitative PCR data. Genome Biol. 2007, 8(2), R19.

Reviewer 2 Report

ABC transporters and okadaic acid production in Prorocentrum lima

Gu et al

This manuscript reports the sequence of three ABC transporters, an analysis of their predicted protein topology, and their expression pattern under different conditions. The cellular levels of okadaic acid were also measured under different conditions.

However, I am unclear as to the connection between the transporters analyzdo ed and okadaic acid. There is no functional information available for the transporters, nor where they are located, so it is difficult to predict what would be the result of increased transporter expression on cellular okadaic acid levels (if they were in the plasma membrane there would be less cellular okadaic acid at higher expression levels, for example). I do not know how many transporters are in the dinoflagellates (there are over a hundred in higher plants) but if there are a lot the chances that the transporters chosen actually transport okadaic acid may not be high. Lastly, even if the transporters chosen were specific for okadaic acid, measures of RNA levels may not be a reliable way to assess altered protein levels or amount of transporter activity.

Line 20            what is tributyltin and why is it used?

Line 134          TBT = ?

Line 162          How many Artemis died?

Line 219          Why is the lack of a signal peptide suggestive of a plasma membrane location?

Line 365          Include phosphate concentration in control (36 uM)

Figure 1 What is the membrane for which the topology of the protein is shown? Is the predicted cytoplasm side the one that binds ATP? Is this the whole derived protein sequence, and if so is there any targeting information? Is the predicted destination destination? Some information is in the discussion (line 219) but some hints at the time when the figure is presented would help.

Figure 3 I understand the different controls in the different panels are all different because they are taken at different times. It would help if this was written in the legend instead of only in the methods. It also seems curious that in some cases the differences are marked as significant because of smaller error bars on one of the samples (panel B for example; is the difference really significant?).

Figure 4 how are the mRNA expression levels reported? I assume they are relative to some value, but I would have expected them to be relative to the control set at 1. Some panels are, but not all. Is there a cut-off value for inferring significance (i.e. greater than two-fold, for example) as is the case for microarrays?

The phylogenies in supplemental figures do not all seem to recover the expected relationships. For example, in Supp. Fig. 3 Symbiodinium and Prorocentrum are quite far apart.

Author Response

Thank you very much for your comments indeed. We have revised the manuscript according to the comments, and carefully proof-read the manuscript to minimize terminological, typographical, grammatical errors. Please see document for details about the revision.

Reviewer 3 Report

The authors present a study describing three ABC transporters in the dinoflagellate P. lima and their potential roles in transport of the phycotoxin okadaic acid (OA) produced by P. lima, and the potential function of OA as an anti-grazing chemical. The introduction is thorough in describing ABC transporter and OA.  However, I find that the work presented does not connect these two aspects.  There is no direct evidence for the role of the described ABC transporters in OA transport, nor even any indirect correlation with their expression in response to OA production.  Therefore, I cannot see why the transporters are being associated with OA, other than that OA was an additional focus of the researchers’ work.   The grammar in the manuscript is poor in several places. Furthermore, the results are not compiled very coherently.

I have a few specific comments below:

100- Figure 1 does not seem to be mentioned in the accompanying results text.

133 - It is not clear why the production of OA was compared under the growth conditions tested. Thus, the context of Fig. 2 is unclear.  Why were the cells cultured in these different conditions?  This section does not explain why the experiments in Fig. 2 and Fig. 3 were done.  The legend of Fig. 2 in particular is poor described and needs much more detail.

161 – what were the culturing conditions for co-cultivation with A salina?  The authors suggest that OA was toxic to A salina.  But no growth data are provided for A salina in the presence/absence of P. lima.

Fig. 4 – y-axis “mRNA expression level”, does this refer to fold change? If so, the fold change is generally very small (2-4 fold).  This accompanying results description are poor for this figure.

Author Response

The authors present a study describing three ABC transporters in the dinoflagellate P. lima and their potential roles in transport of the phycotoxin okadaic acid (OA) produced by P. lima, and the potential function of OA as an anti-grazing chemical. The introduction is thorough in describing ABC transporter and OA. However, I find that the work presented does not connect these two aspects. There is no direct evidence for the role of the described ABC transporters in OA transport, nor even any indirect correlation with their expression in response to OA production. Therefore, I cannot see why the transporters are being associated with OA, other than that OA was an additional focus of the researchers’ work. The grammar in the manuscript is poor in several places. Furthermore, the results are not compiled very coherently.

Answer: Thank you very much indeed for your comments. Indeed, as you commented, we have not found any significant correlation between OA production and expression of the three ABC transporter genes in P. lima, which might be due to the complexity of OA transport in P. lima and mutual influence and interference of other secondary metabolites to transport. On the other hand, these outcomes indicates that ABC proteins might play a limited role in the transport of OA. As for the introduction, to provide a clear formulation for our study, some information has been added in the introduction of the revised manuscript. Finally, we have revised the manuscript according to the reviewers’ comments, and carefully proof-read the manuscript to minimize terminological and grammatical errors.

I have a few specific comments below:

100- Figure 1 does not seem to be mentioned in the accompanying results text.

Answer: I am very sorry for this. Related information has been added in the manuscript.

133 - It is not clear why the production of OA was compared under the growth conditions tested. Thus, the context of Fig. 2 is unclear. Why were the cells cultured in these different conditions? This section does not explain why the experiments in Fig. 2 and Fig. 3 were done. The legend of Fig. 2 in particular is poor described and needs much more detail.

Answer: Given the cellular localization of OA in Prorocentrum lima (Barbier et al., 1999), the differences in sensitivity of OA producing and non-OA producing algae to the deleterious effect of OA (Perreault et al., 2012), and important roles of ABC transporters in the resistance of algae to secondary metabolites and environmental pollutants as important carriers (Kretzschmar et al., 2011), we speculate that ABC transporters might be implicated in transporting or sequestrating endogenous secondary metabolites such as OA and xenobiotic pollutants like heavy metal, and that OA function as an anti-grazing chemical to prevent other organisms grazing. Therefore, our objective are: 1) to provide information on the character of ABC transporter genes in dinoflagellate and their potential functions in adaption of surrounding environment in P. lima; 2) to give some evidence for OA as an anti-grazing chemical; 3) to analyze the potential role of ABC transporters in OA transportation in P. lima. For these, beyond characterization of ABCB1, ABCC1 and ABCG2 in P. lima, OA production in the presence of Artemia salina, a suitable model species to assess the toxicity of marine benthic dinoflagellates, was observed in P. lima. Importantly, given that algal toxin production is found to be regulated by many physiological and ecological factors, such as the imbalance of nutrient ratio, the availability of nutrients, environmental pollutants, and grazing pressure, and so on (Mclachlan et al., 1994; Selander et al., 2008; Dang et al., 2015; Couet et al., 2018), underlying relationship between the three ABC genes expression and OA production under different conditions (nutrient limitations, different sources of nutrients, heavy metal stress, grazing pressure) were analyzed. These are why the experiments in Fig. 2 and Fig. 3 were done. The related descriptions have been added in the introduction and other sections. In addition, more detail information has been added in Figures or Figure legend.

References

Barbier, M.; Amzil, Z.; Mondeguer, F.; Bhaud, Y.; Soyer-Gobillard, M.O.; Lassus, P. Okadaic acid and PP2A cellular immunolocalization in Prorocentrum lima (Dinophyceae). Phycologia 1999, 38(1), 41‒46.

Couet, D.; Pringault, O.; Bancon-Montigny, C.; Briant, N.; Elbaz, F.P.; Delpoux, S.; Kefidaly, O.Y.; Hela, B.; Charaf, M.; Hervé, F.; Rovillon, G., Amzil, Z, Laabir, M. Effects of copper and butyltin compounds on the growth, photosynthetic activity and toxin production of two HAB dinoflagellates: The planktonic Alexandrium catenella and the benthic Ostreopsis cf. ovata. Aquat. Toxicol. 2018, 196, 154‒167.

Dang, L.; Li, Y.; Liu, F.; Zhang, Y.; Yang, W.; Li, H.; Liu, J. Chemical response of the toxic dinoflagellate Karenia mikimotoi against grazing by three species of zooplankton. J. Eukaryot. Microbiol. 2015, 62(4), 470‒480.

Kretzschmar, T.; Burla, B.; Lee, Y.; Martinoia, E.; Nagy, R. Functions of ABC transporters in plants. Essays Biochem. 2011, 50(1), 145–160.

Mclachlan, J.L.; Marr, J.C.; Conlon-Keily, A.; Adamson, A. Effects of nitrogen concentration and cold temperature on DSP-toxin concentrations in the dinoflagellate Prorocentrum lima (prorocentrales, dinophyceae). Nat. Toxins 1994, 2(5), 263–270.

Perreault, F.; Matias, M.S.; Oukarroum, A.; Matias, W.G.; Popovic, R. Okadaic acid inhibits cell growth and photosynthetic electron transport in the alga Dunaliella tertiolecta. Sci. Total. Environ. 2012, 414, 198–204.

Selander, E.; Cervin, G.; Pavia, H. Effects of nitrate and phosphate on grazer-induced toxin production in "Alexandrium minutum". Limnol. Oceanogr. 2008, 53(2), 523–530.

161 – what were the culturing conditions for co-cultivation with A salina? The authors suggest that OA was toxic to A salina. But no growth data are provided for A salina in the presence/absence of P. lima.

Answer: The A. salina individuals were cultured in 2000 mL of seawater sterilized with a salinity of 27 at 20–25 °C for 7 days, and fed with the green alga Tetraselmis subcordiformis. During the experiment, dead A. salina was removed every 24 h and replaced with healthy and lively A. salina individuals. After 48 h, the survival rate of A. salina exposed to P. lima were 72.8%, distinctly lower than their control counterparts (91.1%), suggesting the toxicity of P. lima to the A. salina. The related information has been added in the manuscript.

Fig. 4 – y-axis “mRNA expression level”, does this refer to fold change? If so, the fold change is generally very small (2-4 fold).  This accompanying results description are poor for this figure.

Answer: I apologize for the carelessness in description of mRNA expression levels. They are indeed relative quantity. The software geNorm, NormFinder and Bestkeeper were employed to screen internal genes to normalize expressions of ABCs from the five common housekeeping genes such as 18S RNA, β-actin, β-tubulin, GAPDH and Calm. β-actin and β-tubulin were chosen as internal genes to normalize expression of target genes owing to their most stable expressions. Relative mRNA expression of ABCs gene in different samples was calculated by formul NRQ (Hellemans et al., 2007) with Bio-Rad CFX Mannager 3.0, in which inter-run calibration algorithms were considered. The related descriptions have been added in the revised manuscript. As for the significance, t-test or LSD t-test was employed in different experiments. In A, B, C, F, G, H, I, J, K, and L, asterisks indicate statistically significant differences between control and treatment groups (t-test, * p<0.05). In D and E, bars of respective treatment followed by the same letter are not significantly different at p<0.05 (LSD t-test).

References

Hellemans, J.; Mortier, G.; De Paepe, A.; Speleman, F.; Vandesompele, J. qBase relative quantification framework and software for management and automated analysis of real-time quantitative PCR data. Genome Biol. 2007, 8(2), R19.

Round  2

Reviewer 2 Report

The stated goals of the MS are

1) to provide information on ABC transporters and their potential functions in adapting to the environment

2) provide evidence for OA  as antigrazing

3) analyse potential role of ABC transporters in OA transport

The MS does provide information on three transporters. However, the functions in adapting to the environment are not identified, as transcript abundance increases with N/P limitation, with grazing, and with Cu++ exposure. Presumably the transporters would not provide a functional response to all these different stimuli.

The evidence for anti-grazing is given as Artemis survival in a control experiment (91%) is higher than with P lima (73%). However, it appears this experiment was done once. To make this argument there should be several replicate experiments and the data provided as a mean with a standard deviation.

The authors conclude there is no potential role of these ABC transporters in OA transport based on the observation there is no consistent pattern between transcript levels and OA accumulation. However I am still puzzled by what the authors consider to be the subcellular localisation. The text reports evidence from the literature suggestive of a vacuolar location but the prediction programs used with the derived protein sequences seem to suggest a plasma membrane location.

Author Response

The stated goals of the MS are

1) to provide information on ABC transporters and their potential functions in adapting to the environment

2) provide evidence for OA  as antigrazing

3) analyse potential role of ABC transporters in OA transport 

The MS does provide information on three transporters. However, the functions in adapting to the environment are not identified, as transcript abundance increases with N/P limitation, with grazing, and with Cu++ exposure. Presumably the transporters would not provide a functional response to all these different stimuli.

Answer: Thank you very much indeed for your comments. As you reminded, the three ABC transporters would not provide a functional response to all these different stimuli, though they exhibited various expression profiles under different conditions. High concentration of Cu2+ could up-regulate ABCB1, ABCC1 and ABCG2 transcripts in P. lima, suggesting the potential role of ABC transporters in dinoflagellate defense against metal ions in surrounding waters. As for the alternations under N/P limitation or grazing depression, they might mainly contributed to the physiological responses to stimuli, which concern second metabolites. The three ABC transporters might be regulated by various interrelated metabolic changes as in plant (Bienert et al., 2014). ABC transporters in plant transport a variety of substrates such as lipids, phytohormones, carboxylates, heavy metals, chlorophyll catabolites and xenobiotic conjugates across various biological membranes (Kretzschmar et al., 2011), and have been linked to the response to biotic and/or abiotic stress (Bienert et al., 2014). In previous study, we found that N limitation could induce the accumulation of neutral lipid and starch in P. lima cells for carbon fixation trough recycling chloroplast membranes by autophagy (Hou et al., 2018). For microalgae, carbon storage compounds are often polysaccharides and lipids (Lacour et al., 2012). Interestingly, ABC transporters have been proposed to export polysaccharides outside of dinoflagellate cells (Gong et al., 2017). Accordingly, it seems that ABC transporters might be implicated in transporting of polysaccharides and other second metabolites induced by nutrient limitations. In addition, many studies have demonstrated that grazing pressure could induce harmful algae to produce more toxins and generate other physiological changes in morphology, gene expression and metabolic profile (Amato et al., 2018; Selander et al., 2019). It is likely that alternation of ABC transporter expression might be due to the physiological changes, especially metabolic profile alternation, though we have not detect metabolic profile induced by A. salina. Thank you again for your comment, we have revised some related expression.

References

Amato, A.; Sabatino, V.; Nylund, G.M.; Bergkvist, J.; Basu, S.; Andersson, M.X.; Sanges, R.; Godhe, A.; Kiørboe, T.; Selander, E.; Ferrante, M.I. Grazer-induced transcriptomic and metabolomic response of the chain-forming diatom Skeletonema marinoi. ISME J. 2018, 12, 1594–1604.

Bienert, M.D.; Baijot, A.; Boutry, M. ABCG transporters and their role in the biotic stress response. In: Geisler, M. (eds) Plant ABC Transporters. Signaling and Communication in Plants. Springer, Cham. 2014, 22, 137-162.

Gong, W.; Browne, J.; Hall, N.; Schruth, D.; Paerl, H.; Marchetti, A. Molecular insights into a dinoflagellate bloom. ISME J. 2017, 11, 439–452.

Hou, D.Y.; Mao, X.T.; Gu, S.; Li, H.Y.; Liu, J.S.; Yang, W.D. Systems-level analysis of metabolic mechanism following nitrogen limitation in benthic dinoflagellate Prorocentrum lima. Algal Research 2018, 33, 389–398.

Kretzschmar, T.; Burla, B.; Lee, Y.; Martinoia, E.; Nagy, R. Functions of ABC transporters in plants. Essays Biochem. 2011, 50, 145–160.

Lacour, T.; Sciandra, A.; Talec, A.; Mayzaud, A.; Bernard, O. Neutral lipid and carbohydrate productivities as a response to nitrogen status in Isochrysis sp. (T-ISO; Haptophyceae): starvation versus limitation J. Phycol. 2012, 48, 647–656.

Selander, E.; Berglund, E.C.; Engström, P.; Berggren, F.; Eklund, J.; Harðardóttir, S.; Lundholm, N.; Grebner, W.; Andersson, M.X. Copepods drive large-scale trait-mediated effects in marine plankton. Sci. Adv. 2019, 5, eaat5096.

The evidence for anti-grazing is given as Artemis survival in a control experiment (91%) is higher than with P. lima (73%). However, it appears this experiment was done once. To make this argument there should be several replicate experiments and the data provided as a mean with a standard deviation.

Answer: I apologize for the carelessness in description of survival rate. After 48 h, the survival rate of A. salina exposed to P. lima were 72.8±3.6%, distinctly lower than their control counterparts (91.1±2.5%). The related expression has been added.

The authors conclude there is no potential role of these ABC transporters in OA transport based on the observation there is no consistent pattern between transcript levels and OA accumulation. However I am still puzzled by what the authors consider to be the subcellular localisation. The text reports evidence from the literature suggestive of a vacuolar location but the prediction programs used with the derived protein sequences seem to suggest a plasma membrane location.

Answer: Thank you very much indeed for your comments. According to the previous studies, all the ABCBs in Arabidopsis are localized to the plasma membrane (Hanikenne et al., 2005). Similarly, some of half-size ABCGs in plants are also found localized in the plasma membrane (Chang et al., 2018). Even for the ABCCs, few have been shown to reside on the plasma membrane in plants, despite most are featured as vacuolar localized proteins (Pang et al., 2013). Based on the sequences obtained, we predicted the subcellular localization of the three ABC transporters. It was shown that the three ABC transporters were all located on the cytoplasmic membrane. However, these supposed subcellular localization need more evidence to validate. It is noteworthy that OA, the main DSP toxin, has been proposed to be synthesized in chloroplast, and stored in chloroplast or vacuoles around the cytoplasm. The subcellular localization of the three ABC transporters might diminish the possibility of three transporters transporting OA in P. lima. Nevertheless, DSP toxins-producing dinoflagellate has been shown to secrete toxins into water (Windust et al., 2000), which suggests the underlying role of the three ABC transporters in transporting DSP toxins from the cells. All in all, the production and transportation of DSP toxins are very complicated, and much more studies should be performed. Thank you again for your comment, we have revised some related expression.

References

Chang, Z.Y.; Jin, M.N.; Yan, W.; Chen, H.; Qiu, S.J.; Fu, S.; Xia, J.X.; Liu, Y.C.; Chen, Z.F.; Wu, J.X.; Tang X.Y. The ATP-binding cassette (ABC) transporter OsABCG3 is essential for pollen development in rice. Rice 2018, 11, 58

Hanikenne, M.; Motte, P.; Wu, M.C.S.; Wang, T.; Loppes, R.; Matagne, R.F. A mitochondrial half-size ABC transporter is involved in cadmium tolerance in Chlamydomonas reinhardtii. Plant Cell Environ. 2005, 28(7), 863–873.

Pang, K.; Li, Y.; Liu, M.; Meng, Z.; Yu, Y. Inventory and general analysis of the ATP-binding cassette (ABC) gene superfamily in maize (Zea mays L.). Gene 2013, 526, 411–428.

Windust, A.J.; Hu, T.; Wright, J.L.C.; Quilliam, M.A.; McLachlan, J.L. Oxidative metabolism by Thalassiosira weissflogii (Bacillariophyceae) of a diol-ester of okadaic acid, the diarrhetic shellfish poisoning. J. Phycol. 2000, 36 (2), 342–350.